# Atmospheric Corrosion Sensor Based on Strain Measurement with Active–Dummy Fiber Bragg Grating Sensors

**Nining Purwasih [1,2]**, **Hiroki Shinozaki [2]**, **Shinji Okazaki [3]**, **Hiroshi Kihira [4]**, **Yukihisa Kuriyama [5]** and **Naoya Kasai [2,*]**

[1]   Electrical Engineering, Lampung University, Lampung 35145, Indonesia; nining.purwasih@eng.unila.ac.id
[2]   Graduate School of Environment and Information Sciences, Yokohama National University, Yokohama 240-8501, Japan; shinozaki-hiroki@fujielectric.com
[3]   Graduate School of Engineering, Yokohama National University, Yokohama 240-8501, Japan; okazaki-shinji-yp@ynu.ac.jp
[4]   Nippon Steel & Sumikin Research Institute, Tokyo 100-0005, Japan; kihira.hiroshi@nsri.nssmc.com
[5]   RACE Research into Artifacts, Center for Engineering, University of Tokyo, Tokyo 113-0033, Japan; kuriyama@race.u-tokyo.ac.jp
*   Correspondence: kasai-naoya-pf@ynu.ac.jp; Tel.: +81-45-339-3979

**Abstract:** Using the relationship between strain and thickness from the materials theory, this paper presents further development of the atmospheric corrosion sensor based on strain measurement (ACSSM). Fiber Bragg grating (FBG) sensors were used to measure strain in this study. The active–dummy method was employed to compensate the effects of environmental temperature drift, with the configuration and position of the active–dummy FBG sensors determined based on simulations conducted using the finite element method (FEM). Hence, the reduction in thickness of low-carbon steel test pieces could be isolated, ensuring accurate characterization of this parameter. Results of practical galvanostatic electrolysis experiments conducted with the FBG sensors in the proposed configuration demonstrated accurate measurement of the reduction in the thickness of a test piece, suggesting that an ACSSM with active–dummy FBG sensors would be appropriate for monitoring of atmospheric corrosion in steel structures.

**Keywords:** atmospheric corrosion sensor; strain measurement; FBG sensors; low-carbon steel

---

## 1. Introduction

Since corrosion is the main failure mode for steel structures, sensors that can predict their corrosion rate are very important in structural health monitoring for evaluating safety levels [1]. The detection of losses in weight and thickness [2,3] and electrochemical impedance spectroscopy [4,5] are techniques that have been proposed for atmospheric corrosion monitoring. Although the former methods are capable of measurement to a high degree of accuracy, they do not provide real time monitoring, and the cost of chemicals required for removing the corrosion product is high. Similarly, while electrochemical methods allow in situ corrosion monitoring, a precise measurement is difficult as the sensitivity of these techniques to corrosion reactions affects their operation. Thus, a highly accurate in situ sensor capable of monitoring atmospheric corrosion is needed. Based on this requirement, we developed the atmospheric corrosion sensor based on strain measurement (ACSSM).

In previous ACSSM studies [6–8], the reduction of the thickness of a test piece due to corrosion was evaluated using elastic strain by applying a bending moment to the test piece. The mechanical principles of the changes to elastic strain due to reductions in thickness produced by corrosion were

devised from theoretical formulas and studies of finite element method (FEM) simulations, using galvanostatic electrolysis as a model for corrosion. Although the actual strain to be measured for evaluation of the reduction of thickness is very small, for long monitoring periods, the strain gauge is subject to large amounts of noise due to changes in the environmental temperature. Hence, we fabricated an original strain measurement circuit with active–dummy circuits for the ACSSM [7,8], which obtained results in good agreement with the mechanical principles devised. However, changes in strain due to a reduction in thickness were very small, and there was a 12% difference between the thicknesses estimated using strain and those obtained by measuring the actual dimensions and the weight loss of the test piece.

In the last decade, many optical fiber techniques have been used such as distributed fiber sensors [9,10], low coherent optical fiber interferometer [11,12], and long period grating sensors [13,14] for strain measurements including fiber Bragg grating (FBG) sensors that were used in this research. FBG has been used in many structural healthy monitoring applications, such as monitoring strain in a metal bridge [15], strain in a concrete structure [16], the stretching of cables in a gymnasium structure [17], the displacement of a landslide [18], detecting the welding joints in a structure [19], and monitoring strain in material for aircraft [20]. Several FBG-based methods, such as FBG sensors embedded in metal film coatings [21,22] and sensors with etched metal cladding for increased sensitivity [23,24], have been proposed for corrosion monitoring in steel structures. Moreover, such sensors have exhibited good sensitivity to the progression of corrosion, even when embedded in concrete [25–27]. Since strain measurements using FBG sensors are accurate, of high resolution, and stable, we have modified our ACSSM technique to include active–dummy FBG sensors. In the study, to accurately evaluate the reduction in the thickness of the test piece, the configuration and install positions of the active and dummy FBG sensors were determined through FEM analysis of the effect of temperature on the strain measurement. In addition, we verified the accuracy of the active–dummy FGB sensors' estimation of reductions in thickness due to corrosion using laboratory experiments based on galvanostatic electrolysis [28].

The operating principle of the ACSSM technique is the measurement of the reduction in thickness of a test piece using the mechanical theory of deformation. An illustration of how this theory is applied is shown in Figure 1. Here, a test piece with a thickness $h$ is deformed elastically according to the radius of curvature, $\rho$, and the center angle, $\mathrm{d}\theta$. The neutral plane (N-N) is unaltered by the deformation. The shortened curvature in the compressed strain position (C-C) where the FBG sensor is installed is equal to $\left(\rho - \frac{h}{2}\right)\mathrm{d}\theta$. The strain on the compressed surface can be expressed by [6,7]:

$$\varepsilon = \frac{\left(\rho - \frac{h}{2}\right)\mathrm{d}\theta - \rho\mathrm{d}\theta}{\rho\mathrm{d}\theta} = -\frac{h}{2\rho} \tag{1}$$

If the thickness of the test piece is decreased due to corrosion, the distance between the neutral plane and the surface under the corroded area is decreased, as shown in Figure 1b. From (1), assuming that $\rho >> h$, the change in strain can be expressed as

$$\Delta h = 2\rho \cdot \Delta \varepsilon \tag{2}$$

Hence, according to the above, for a constant value of $\rho$, the change in the strain of the concave surface determines the change in the thickness of the test piece. To ensure that the test piece is subject only to elastic deformation, the minimum value of $\rho$ is calculated as follows [6]:

$$\rho = \frac{Eh}{2\sigma} \tag{3}$$

where $\sigma$ is the yield stress of the test piece, and $E$ is its Young's modulus.

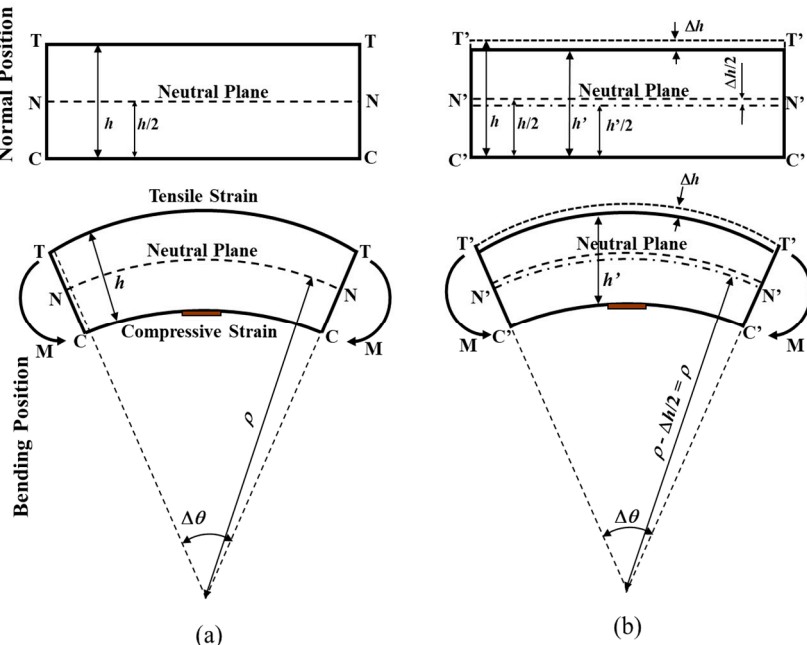

**Figure 1.** Illustration of (**a**) a non-corroded test piece, and (**b**) a corroded test piece highlighting the geometry of the normal and bending positions.

The operation of the FBG strain sensor is based on the measurement of the shift in the Bragg wavelength, which is caused by changes in strain, temperature, and other external influences. In unstrained conditions, the Bragg wavelength ($\lambda_B$) is given by [23,25]:

$$\lambda_B = 2 \cdot n \cdot \Lambda \tag{4}$$

where $n$ is the refractive index and $\Lambda$ is the spacing between the gratings. When there is a force and temperature from the external environment, both the refractive index and the spacing between the gratings change, and the Bragg wavelength is shifted accordingly:

$$\lambda_B = 2 \cdot n \cdot \Delta\Lambda + 2\Delta n \cdot \Lambda \tag{5}$$

where $\Delta\Lambda$ is the change in spacing between gratings and $\Delta n$ is the change in the refractive index. The changes in the Bragg wavelength due to temperature and strain can be determined by using (5), where the first and second term represent the changes due to strain and temperature, respectively, in [29]:

$$\Delta\lambda_B = [k\varepsilon + (\alpha_\Lambda + \alpha_n)\Delta T]\lambda_B \tag{6}$$

where $k$ is a gauge factor, $\alpha_\Lambda$ is the thermal expansion coefficient, $\alpha_n$ is the thermo-optic coefficient, and $\Delta T$ is the change in temperature.

## 2. Materials and Methods

### 2.1. Experimental Apparatus

Figure 2 shows a diagram of the experimental apparatus, which comprises a base and cover made of polyvinyl chloride. The test piece was 95 mm in length, 45 mm in width, and 0.5 mm in thickness. A 1350 mm$^2$ area (30 mm × 45 mm) of the test piece was exposed to facilitate the corrosion representative of the reaction occurring in the rest of the steel structure. The reduction in the thickness of this corroded area enabled verification of the detection principle. The FBGs were installed in the back side of the test piece in the apparatus. The low-carbon material used for the test piece has a Young's modulus of 210

GPa and a yield stress of 240 MPa. Hence, the minimum value of $\rho$ can be determined to be ~218.75 mm, using (3). To prevent local plastic deformation, the value of $\rho$ was designed to be 430 mm, ~2 times as large as the minimum.

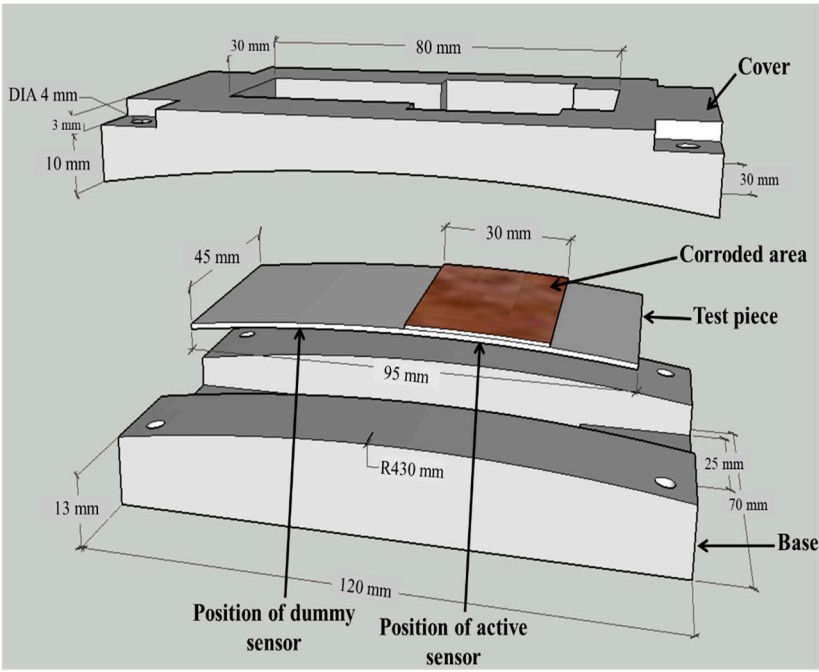

**Figure 2.** Diagram of the atmospheric corrosion sensor based on strain measurement (ACSSM) apparatus, consisting of a base, the test piece, and a cover.

## 2.2. Measurement Using the Active–Dummy Method

The active–dummy method employed with the ACSSM requires the use of two FBG sensors (an active sensor and a dummy sensor) with different functions. $\varepsilon_A$ and $\varepsilon_D$, respectively, the outputs of the active and dummy FBG sensors in $\mu\varepsilon$ can be expressed as follows:

$$\varepsilon_A = \left(\frac{\Delta\lambda_{BA}}{k\lambda_{BA}}\right) - \frac{1}{k}(\alpha_\Lambda + \alpha_n)\Delta T \tag{7}$$

$$\varepsilon_D = \left(\frac{\Delta\lambda_{BD}}{k\lambda_{BD}}\right) - \frac{1}{k}(\alpha_\Lambda + \alpha_n)\Delta T \tag{8}$$

where $\lambda_{BA}$ and $\lambda_{BD}$ are the initial wavelengths, and $\Delta\lambda_{BA}$ and $\Delta\lambda_{BD}$ are changes in the wavelength of the active and dummy FBG sensors, respectively, in nm. Therefore, $\Delta\varepsilon_{AD}$, the difference between the strain in both sensors in $\mu\varepsilon$, is expressed as follows:

$$\Delta\varepsilon_{AD} = \varepsilon_A - \varepsilon_D = \left(\frac{\Delta\lambda_{BA}}{k\lambda_{BA}}\right) - \left(\frac{\Delta\lambda_{BD}}{k\lambda_{BD}}\right) \tag{9}$$

Inspection of (9) highlights that with the active–dummy method, the dependence of the output of both sensors on $\Delta T$ has been removed. Hence, unlike the systems described in [30–33], no additional temperature sensor is required to calculate strain. Moreover, with appropriate configuration, $\varepsilon_A$ characterizes both the strain due to the reduction in the test piece's thickness and strain due to environmental factors, while $\varepsilon_D$ characterizes only the strain due to environmental factors. Hence, using $\Delta\varepsilon_{AD}$, the strain from reductions in thickness due to corrosion can be isolated, enabling more accurate measurement.

### 2.3. Configuration of FBG Sensors on the Test Piece

Two os3200 FBG sensors from Micron Optic with initial wavelengths of 1548 nm and $k = 0.796$ were used as active and dummy sensors in experiments to monitor the strain on the test piece. To verify the operation of the active–dummy method, an additional os4200 FBG sensor from Micron Optic with an initial wavelength $\lambda$ of 1556 nm was used as a temperature sensor.

Figure 3 shows the configuration of the FBG sensors on the test piece. Sensors were placed on the reverse of the corroded area of the test piece (Figure 3a) in the configuration shown in Figure 3b. The active FBG sensor was placed parallel to the longitudinal axis, while the dummy FBG sensor was placed parallel to the transverse direction of the test piece in the compressed curve apparatus. An actual figure of the FBG sensors in the test piece is shown in Figure 3c.

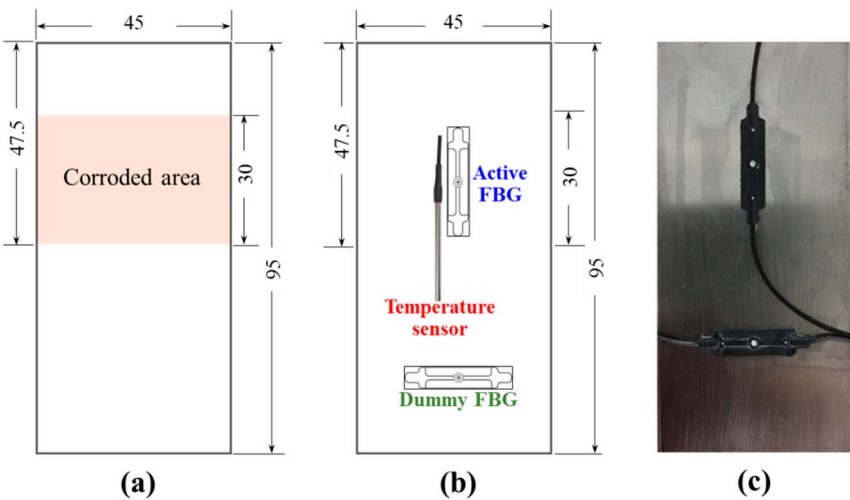

**Figure 3.** Configuration of active and dummy fiber Bragg grating (FBG) sensor on the (**a**) front side and (**b**) back side of the test piece. (**c**) The actual figure of the FBG sensors attached to the test piece. The unit is in mm.

### 2.4. Compensation of Thermal Strain

We firstly conducted preliminary experiments to investigate the effect of environmental temperature drift on the strain measured by the FBG sensors of the ACSSM. Strain and temperature reflection spectra were measured with a commercial optical interrogation unit from Micron Optics (optical sensing interrogator, model sm125) for 88 h with a data sampling interval of 1 min, controlled by the ENLIGHT software package. As mentioned above, two os3200 FBG sensors from Micron Optics were used as active and dummy strain sensors and an os4200 FBG sensor was used as a temperature sensor. The data obtained from this test were used for further calibration in subsequent experiments.

### 2.5. Accelerated Corrosion Using Galvanostatic Electrolysis

We conducted galvanostatic electrolysis experiments to verify the accuracy of our technique in identifying corrosion rates. An illustration of the set-up used in these experiments is shown in Figure 4. We used a 1 mol/L hydrochloric acid solution for accelerated simulation of the effects of environmental corrosion, and a stirrer to ensure a homogeneous liquid mixture. To reduce its thickness, a current of 0.3 A was applied to the test piece, which acted as the working electrode, through the potentiostat/galvanostat. The strain on the test piece was measured using an optical sensing interrogator and personal computer setup sampling at an interval of 30 s. The thickness of the test piece was measured before and after the experiment to verify the change in thickness ($\Delta h$) estimated from the strain measurement. Finally, the working electrode and counter electrode were constructed from the same material.

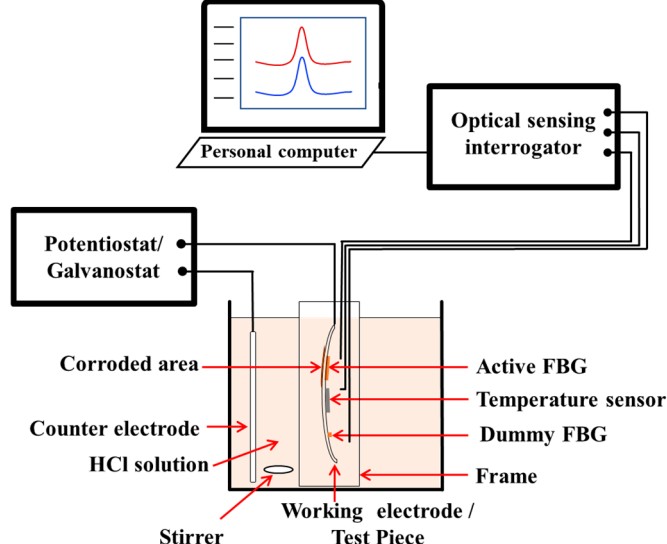

**Figure 4.** Set-up of galvanostatic electrolysis experiments to reduce thickness of the test piece.

## 3. Result

### 3.1. Numerical Analysis

To isolate components of strain resulting from reduced test piece thickness due to corrosion from components resulting from deviations to the experimental environment, accurate configuration of the positions of the active and dummy FBG sensors is necessary. To determine these positions, we analyzed the behavior of the low-carbon steel test pieces subject to various sources of strain with FEM, using a commercial software (ANSYS Mechanical APDL 18.2 from ANSYS Inc., (Canonsburg, PA, USA). Figure 5 shows the analytical geometry and boundary conditions applied to the test piece before applying $\rho$ of 430 mm to the test piece. The $z$ displacement along the $y$-axis at the $x = 0$ node of the test piece was fixed to 0 before/after applying $\rho$ of 430 mm to the test piece. After applying $\rho$ of 430 mm to the test piece, the $z$ coordinates of the test piece at each $x$ coordinates were determined according to the equation below:

$$z = -430 + 2.6315 - \sqrt{430^2 - (x - 47.5)^2} \tag{10}$$

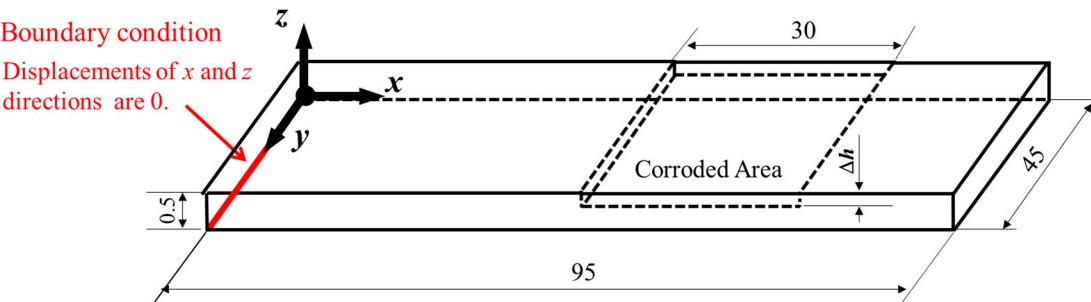

**Figure 5.** Model of the test piece and boundary conditions used in simulation. The unit is in mm.

The geometry of the experimental apparatus indicates that the back side of the test piece experiences a compressive strain, while the surface of the piece experiences a tensile strain (Figure 1). Figure 6 shows the analytical result with a thickness reduction of 0.2 mm. Red and yellow colors mean 581 $\mu\varepsilon$ and 350 $\mu\varepsilon$ in axial direction. The dimension is 95 mm in length and 45 mm in width and the corroded area is 1350 mm$^2$ (30 mm × 45 mm). Figure 7 shows the side view of the $xy$ plane of simulated strain distribution on the back side of the test piece with and without a reduction to the thickness of the

corroded area, with Figure 7a depicting the distribution in the axial direction, and Figure 7b depicting the distribution in the transverse direction. Without a reduction in thickness ($h = 0.5$ mm), a uniform strain (as indicated by the homogenous blue shading) of $-579$ $\mu\varepsilon$ was calculated in the axial direction from an average of five points, a value which is similar to the one predicted by analytical methods ($-581$ $\mu\varepsilon$). In contrast, in the transverse direction, the strain was 0 $\mu\varepsilon$, indicating that the moment applied a longitudinal strain in the $x$ direction of the test piece. Similarly, for the corroded area of the test piece, when $h = 0.1$ mm and $h = 0.27$ mm, the magnitude of the strain in the axial direction predicted using numerical and analytical models was the same. Based on these FEM simulations, a decrease in the thickness of the corroded area of the test piece does not affect the strain distribution in the transverse direction.

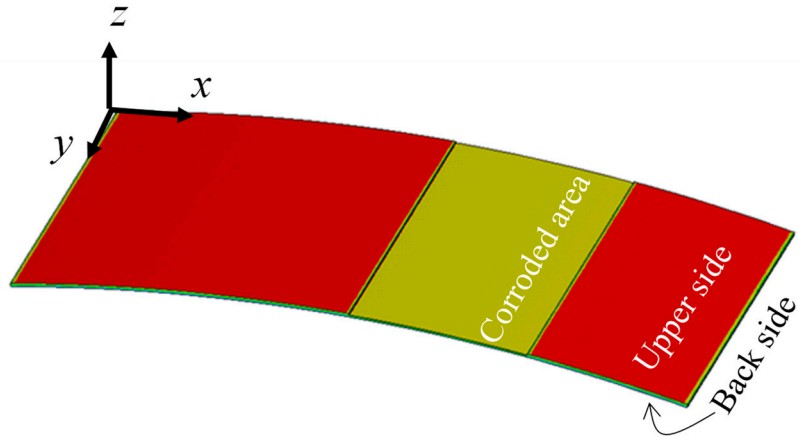

**Figure 6.** Analytical result with a thickness reduction of 0.2 mm. Red and yellow colors mean 581 $\mu\varepsilon$ and 350 $\mu\varepsilon$ in axial direction. The dimension is 95 mm in length and 45 mm in width and the corroded area is 1350 mm$^2$ (30 mm × 45 mm).

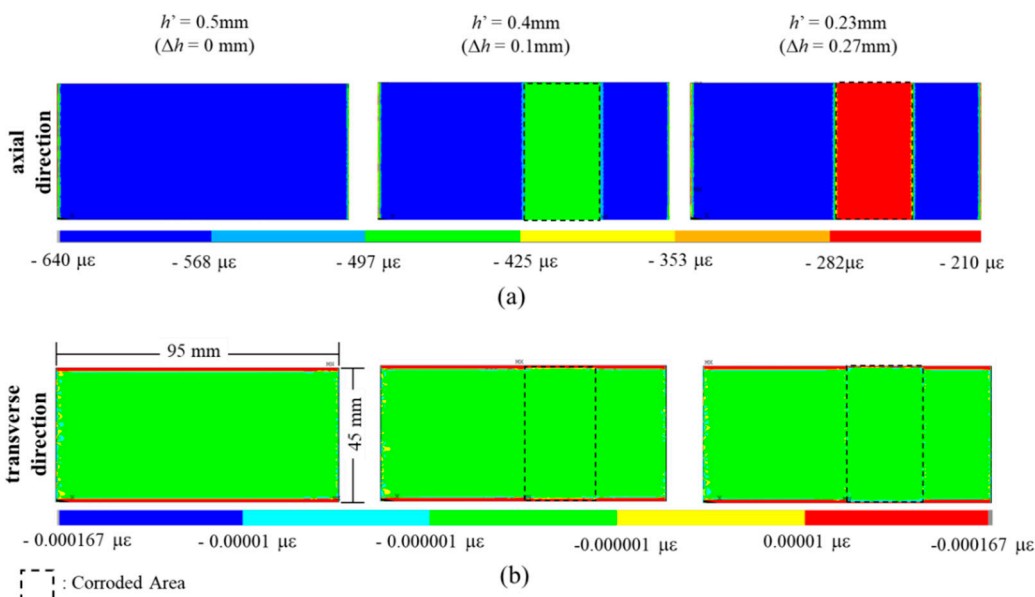

**Figure 7.** The side view of $xy$ plane of FEM-simulated strain distribution on the back side of the test piece with and without a reduction in thickness, (**a**) in the axial direction, and (**b**) in the transverse direction.

The relationship between the longitudinal strain and the thickness of the test piece is shown in Figure 8; Figure 8a shows the relationship between the absolute strain in the $x$ direction and the thickness of the test piece, while Figure 8b shows the relationship between the change in strain ($\Delta\varepsilon$) and

change in thickness ($\Delta h$). We note a linear relationship between these two parameters, as suggested by (2), and good correlation between the analytical results and the finite element simulation, indicating that the operating principle of the atmospheric corrosion sensor is reasonable.

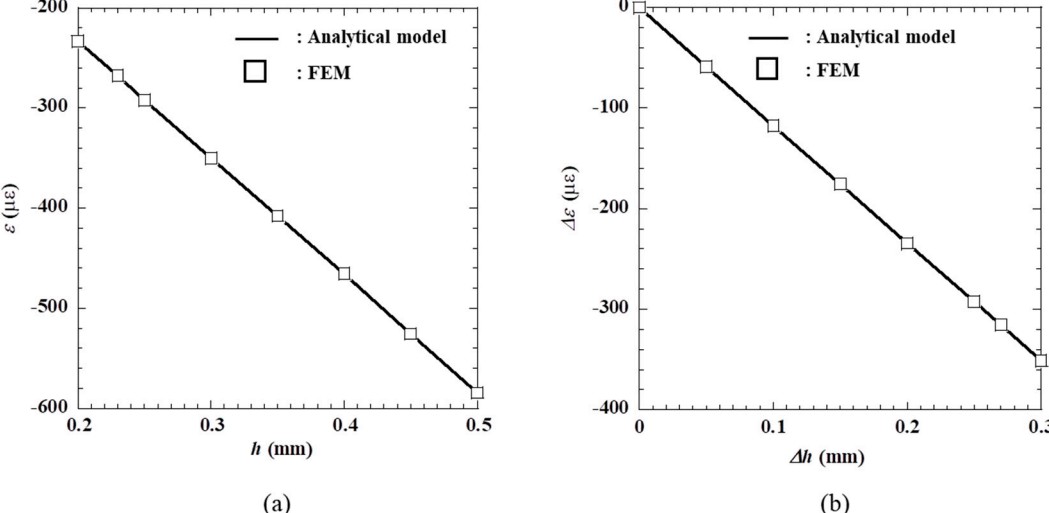

(a)                                                                        (b)

**Figure 8.** (**a**) Relationship between the strain ($\varepsilon$) and thickness of the test piece ($h$) from FEM simulations. (**b**) Relationship between the change in strain ($\Delta\varepsilon$) and change in thickness $\Delta h$ in mm from FEM simulations. The analytical model provides the line of best fit in both plots.

A summary of the results of FEM simulations (conducted at a base temperature of 300 K and an elevated temperature of 310 K) is given in Table 1. Here, we have included the strain in the axial direction in the corroded area ($\varepsilon^{C}$) where the active FBG sensor was set up, and the strain in the axial and transverse directions under the uncorroded area ($\varepsilon^{UC}$) where the dummy FBG was set up. $\Delta\varepsilon^{C}$ was obtained from the strain after thickness reduction ($\varepsilon^{C}_{\Delta h} - \varepsilon^{C}_{\Delta h=0}$). When $\Delta T = 10$ K, the strain changed by 117 $\mu\varepsilon$ in both the axial and transverse directions, corresponding to a uniform thermal expansion of 11.7 $\mu\varepsilon$/K. The results indicate that the thermal strain in the corroded and uncorroded areas is the same, in both the axial and transverse directions.

**Table 1.** Strains obtained from FEM simulation.

| | | Strain ($\mu\varepsilon$) | | | | | | | | | | |
| --- | --- | --- | --- | --- | --- | --- | --- | --- | --- | --- | --- | --- |
| | | Corroded Area | | | | | Uncorroded Area | | | | | |
| $h$ [mm] | $\Delta h$ [mm] | Axial Direction | | | | | Axial Direction | | | Transverse Direction | | |
| | | $\varepsilon^{C}_{\Delta h}$ at 300 K | $\varepsilon^{C}$ at 300 K | $\varepsilon^{C}_{\Delta h}$ at 310 K | $\Delta\varepsilon^{C}$ at 310 K | $\Delta\varepsilon^{C}_{\Delta T=10\,K}$ | $\Delta\varepsilon^{UC}$ at 300 K | $\Delta\varepsilon^{UC}$ at 310 K | $\Delta\varepsilon^{UC}_{\Delta T=10\,K}$ | $\Delta\varepsilon^{UC}$ at 300 K | $\Delta\varepsilon^{UC}$ at 310 K | $\Delta\varepsilon^{UC}_{\Delta T=10\,K}$ |
| 0.5 | 0 | −584 | 0 | −467 | 0 | 117 | −584 | −467 | 117 | 0 | 117 | 117 |
| 0.45 | 0.05 | −525 | 59 | −408 | 59 | 117 | | | | | | |
| 0.4 | 0.1 | −466 | 118 | −349 | 118 | 117 | | | | | | |
| 0.35 | 0.15 | −408 | 176 | −291 | 176 | 117 | | | - | | | |
| 0.3 | 0.2 | −350 | 234 | −233 | 234 | 117 | | | | | | |
| 0.25 | 0.25 | −292 | 292 | −175 | 292 | 117 | | | | | | |
| 0.23 | 0.27 | −268 | 316 | −151 | 316 | 117 | | | | | | |
| 0.2 | 0.3 | −233 | 351 | −116 | 351 | 117 | | | | | | |

By using Equations (7)–(9), $\varepsilon_{A}$, $\varepsilon_{D}$, and $\Delta\varepsilon_{AD}$ can be written as follows:

$$\varepsilon_{A} = \Delta\varepsilon^{C} - \Delta\varepsilon^{C}_{\Delta T} \tag{11}$$

$$\varepsilon_{D} = \Delta\varepsilon^{UC} - \Delta\varepsilon^{UC}_{\Delta T} \tag{12}$$

$$\Delta\varepsilon_{AD} = \varepsilon_{A} - \varepsilon_{D} = \Delta\varepsilon^{C} - \Delta\varepsilon^{C}_{\Delta T} - \left(\Delta\varepsilon^{UC} - \Delta\varepsilon^{UC}_{\Delta T}\right) \tag{13}$$

From FEM simulation in Table 1, $\Delta\varepsilon_{\Delta T}^{C} = \Delta\varepsilon_{\Delta T}^{UC} = 117\ \mu\varepsilon$ for $\Delta T = 10$ K. Thus,

$$\Delta\varepsilon_{AD} = \Delta\varepsilon^{C} - \Delta\varepsilon^{UC} \tag{14}$$

$\Delta\varepsilon^{C}$ and $\Delta\varepsilon_{\Delta T}^{C}$ are the differential strains in the corroded area which correspond with the first and second term in Equation (7), respectively. $\Delta\varepsilon^{UC}$ and $\Delta\varepsilon_{\Delta T}^{UC}$ are the difference strains in the uncorroded area which correspond with the first and second term in Equation (8). $\Delta\varepsilon_{AD}$ is the purposed signal which has relation with the thickness reduction of the test piece, as shown in Equation (2). We used this parameter for the experimental parameter as describes in the Results and Discussion sections.

The side view of the *xy* plane of the magnified strain distribution in the axial direction of a test piece with a thickness reduction of 0.27 mm is shown in Figure 9. We note a 350 µε difference between the strain at the edge and the center of the corroded area. More importantly, the strain is not uniform in the uncorroded area; a difference of approximately 12 µε can be observed. This non-uniform strain affects accurate estimation of the reduction in the thickness of the test piece, since the changes to be measured are very small. Hence, we set up the dummy FBG sensor in the *y* direction of the uncorroded area, as the strain due to environmental factors is identical in both directions, explaining the configuration of the active and dummy FBG sensors on the test piece shown in Figure 3.

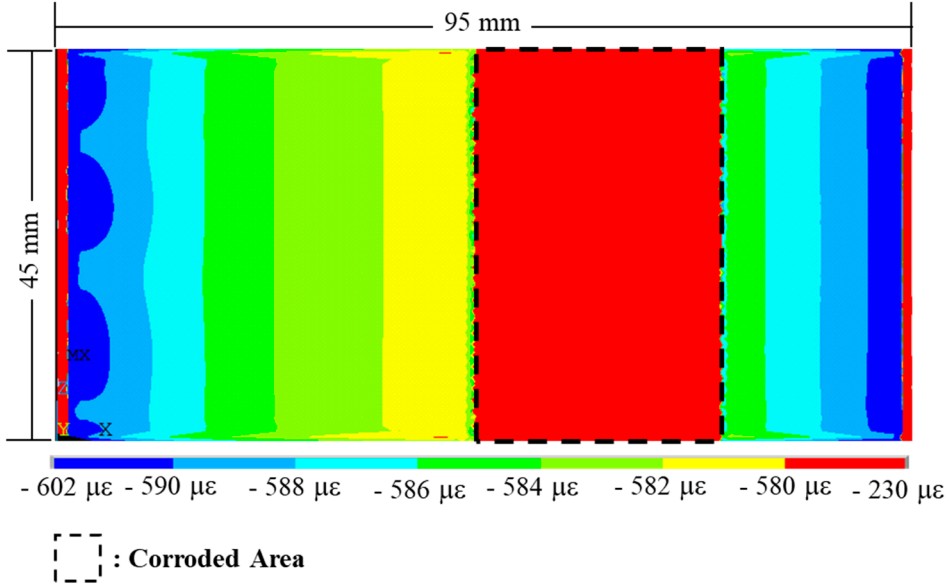

**Figure 9.** The side view of the *xy* plane of magnified strain distribution on the back side of the test piece in the axial direction, with a 0.27 mm reduction in thickness.

### 3.2. Thermal Strain Compensation

Figure 10 shows the results of the thermal strain measured by the FBG sensors of the ACSSM in the preliminary experiments. In this figure, $T_{TP}$ is the temperature of the test piece in degrees centigrade, $\varepsilon_{A}$ is the strain from the active FBG sensor in µε, $\varepsilon_{A}$ is the strain from the dummy FBG sensor in µε, and $\Delta\varepsilon_{AD}$ is the difference in strain between $\varepsilon_{A}$ and $\varepsilon_{D}$. Both sensors had similar responses to temperature variations, producing a maximum change in strain of approximately 52 µε for a 2.8 °C temperature variation, in the absence of changes to the thickness of the test piece. In contrast, for the same variation in temperature, the maximum change in the value of $\Delta\varepsilon_{AD}$ was ~10 µε. This reduced sensitivity to temperature makes this signal more suitable for strain measurement to determine the atmospheric corrosion rate, which requires high resolution and long-term data monitoring in the field.

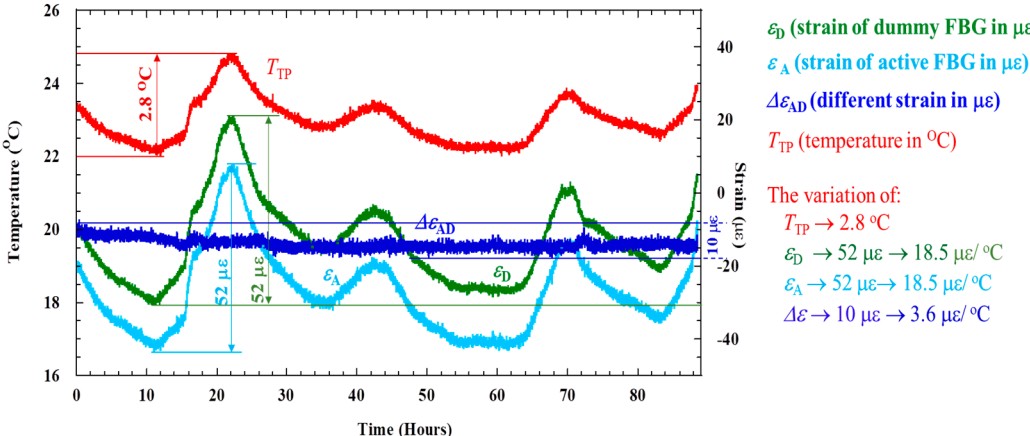

**Figure 10.** Variation in the sensor signal patterns in the absence of corrosion. Only $\Delta\varepsilon_{AD}$ shows no dependence on temperature.

### 3.3. Galvano-Static Electrolysis Experiment

Figure 11 shows the results of the galvanostatic electrolysis experiment, with the light blue line representing the output of the active strain sensor ($\varepsilon_A$), the green line representing the output of the dummy strain sensor ($\varepsilon_D$), and the red line representing the output of the temperature sensor ($T_{Tp}$). All three sensors produced a similar output before the current was applied from the potensiostat/galvanostat. In contrast, on the application of the current, a linear increase in the strain signal could be observed during electrolysis using $\varepsilon_A$. $\varepsilon_D$ and $T_{Tp}$ exhibit a similar behavior. The strain signals remain fairly static, with both sensors indicating that the electrolysis process does not result in extreme temperature changes. The dark blue line in this figure is $\Delta\varepsilon_{AD}$, which exhibits a similar trend to $\varepsilon_A$.

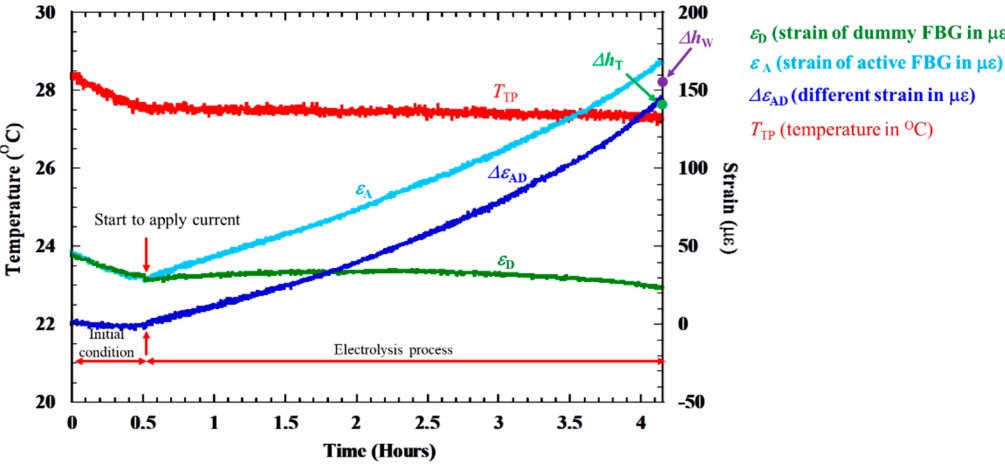

**Figure 11.** Strains measured during the electrolysis process. Equivalent strain from the actual change in thickness ($\Delta h_T$) and change in thickness evaluated from weight loss ($\Delta h_W$) are labeled on the plot for comparison.

## 4. Discussion

A figure of the test piece following the experiment is shown in Figure 12a, which depicts the effects of electrolysis on the apparatus. To evaluate the accuracy of the active–dummy FBG sensors' estimation of changes in thickness, we measured the actual thickness and weight of the test piece before and after electrolysis. The labeled points in Figure 12b were measured using a micrometer, and the thickness of the test piece was determined based on the average of these measurements. The average values calculated before and after corrosion were 0.48 mm and 0.36 mm, respectively, corresponding to

a change in thickness ($\Delta h_T$) of 0.12 mm. To estimate change in thickness using the weight of the test piece, we evaluated the equation below:

$$\Delta h_W = \frac{\Delta W}{S\,d} \tag{15}$$

where $\Delta W$ is the change in weight, $S$ is the area of the corroded region (1350 mm$^2$), and $d$ is the density of the test piece (0.0078 g·mm$^{-3}$). The test piece was weighed at 16.0 g and 14.6 g before and after the experiment, respectively. Hence, $\Delta W = 1.4$ g, and from Equation (15), $\Delta h_W = 0.132$ mm. The final value of $\Delta\varepsilon_{AD}$ obtained in the electrolysis experiment was 143 µε, indicating that $\Delta h = 0.12$ mm, according to (2). A comparison of these results is given in Table 2. A maximum difference in thickness of 6.8% was noted, highlighting that the modifications have improved the performance of the ACSSM, which produced a difference of 12% in previous experiments.

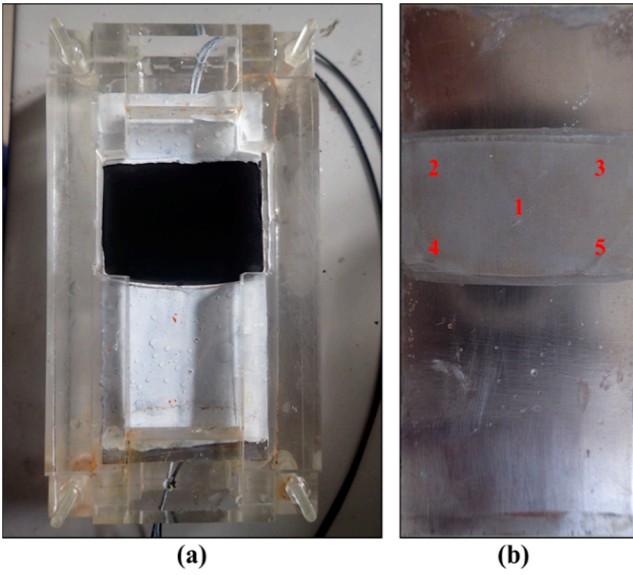

**Figure 12.** (**a**) Test piece in the experimental apparatus following electrolysis. (**b**) Location of points measured for evaluation of change in thickness after electrolysis.

**Table 2.** Comparison of thickness reduction measured by thickness and weight loss.

|  | Based on Strain Measurement ($\Delta h$) | Based on Actual Thickness ($\Delta h_T$) | Based on Weight Loss ($\Delta h_W$) |
|---|---|---|---|
| Thickness (µm) | 123 | 120 | 132 |
| Difference (%) | - | 2.5 | 6.8 |

## 5. Conclusions

In this study, we proposed an improvement to the ACSSM technique, replacing conventional strain gauges with active–dummy FBG sensors. The orthogonal configuration of active–dummy FBG sensors on the test piece was determined from FEM analysis of the thermal expansion and the reduction in the thickness of the test piece. We observed that $\Delta\varepsilon$, the difference between the outputs of the active and the dummy FBG sensors, is less sensitive to changes to the experimental environment, enabling the collection of more accurate and constant signals less affected by temperature drift during prolonged measurement. Results of galvanostatic electrolysis experiments indicated an approximate difference of 6.8% between thicknesses estimated using the strain measured with the active–dummy FBG sensors and that estimated from weight loss, and 2.5% for actual thicknesses measured using a digital micrometer. These small differences suggest that $\Delta\varepsilon$ can be used to determine the atmospheric corrosion rate, which requires high resolution in the field. This quality, in addition to the ability to

perform long-term data monitoring, makes the ACSSM with FBG sensors suitable for estimating the atmospheric corrosion of steel structures. Monitoring a test piece with the ACSSM that is fabricated from the same material as the steel structure and exposed to the same environment will contribute to effective decision-making for the maintenance of the steel structure.

**Author Contributions:** Developed the theory and proofs for the outline, N.K., H.K. and S.O.; encouraged, N.K.; Investigated and supervised the findings of this work, N.P. and H.S.; designed the model, performed the experiments and analyzed the data, N.P., N.K., and S.O.; carried out the FBG experiment, N.P.; and performed the FEM simulation, S.H. All authors discussed the results and contributed to the final manuscript. All authors have read and agreed to the published version of the manuscript.

**Funding:** This research was supported by Grant-in-Aid for Scientific Research (B) JSPS KAKENHI Grant No. 16H03132 and Yokohama National University.

**Acknowledgments:** The authors would like to thank the Institute of Advanced Sciences and the Graduate School of Environment and Information Sciences of Yokohama National University and the Indonesian Directorate General of Higher Education for their financial support of the author's study.

**Conflicts of Interest:** The authors declare no conflict of interest.

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
