# Peer review of "Atmospheric Corrosion Sensor Based on Strain Measurement with Active–Dummy Fiber Bragg Grating Sensors"

_metals, doi:10.3390/met10081076_

Round 1
Reviewer 1 Report
Manuscript is written and presented well with enough technical discussion for publication. But some clarification required before publication.
- In Fig.7b, the starting and ending scale bar is same with different color code. Why?
- Will the FBG sensors work for strain measurement on a very thick metal sheet/structures instead of thin metal sheet used for this study?
Author Response
Response to Reviewer 1 Comments
Manuscript is written and presented well with enough technical discussion for publication. But some clarification required before publication.
Point 1: In Fig.7b, the starting and ending scale bar is same with different color code. Why?
Response 1: Because we would like to indicate that the strain of transversal direction (y-direction) in the all area test piece has very small changes which are the results of Fig. 7(b). It means that the strain of transversal direction is not affected by bending moment and corrosion. Based on the results, the dummy FBG sensor direction which has not affected by corrosion was determined.
Point 2: Will the FBG sensors work for strain measurement on very thick metal sheets/structures instead of a thin metal sheet used for this study?
Response 2: It depends on the thickness of the metal sheet. It is because we have to and make the curvature of a test piece by applying bending moment in our study, and a thicker metal sheet is difficult to make the curvature of a test piece by bending moment.

Reviewer 2 Report
In my opinion, the manuscript should be published as it is. I have found the high importance and novelty in this contribution. I am impressed with the content.
Thank you for the opportunity to evaluate this manuscript.
Author Response
Response to Reviewer 2 Comments
Point 1: In my opinion, the manuscript should be published as it is. I have found the high importance and novelty in this contribution. I am impressed with the content. Thank you for the opportunity to evaluate this manuscript.
Response 1: Thank you very much for your comment to encourage us.

Reviewer 3 Report
In this manuscript, Purwasih et al. presents the development of an atmospheric corrosion sensor based on strain measurement. The sensor has been constructed by replacing conventional strain gauges with Fiber Bragg grating sensors and by applying the active-dummy method to compensate effects of environmental temperature. The steps of fabricating the sensor and testing sensor’s performance is well-described. Results deserve publication. Authors should take into account the following comments while preparing the final version of the manuscript:
--- I have not received the graphical abstract. If authors have not prepared graphical abstract yet, it should be submitted together with the final version of the manuscript.
--- References are not following the required ACS style guide. Authors must correct references accordingly (see journal home page), e.g. Journal Articles:
1. Author 1, A.B.; Author 2, C.D. Title of the article. Abbreviated Journal Name Year, Volume, page range.
--- Line 196: replace “Fig. 6” with “Fig. 7”
--- The English of the manuscript has to be improved. A final reading is required to remove typing mistakes and correct wrong sentence structures. I name only one example: “Fabricating the test piece from the same material as the steel structures it is monitoring enables the ACSSM to contribute to effective decision making for their maintenance.” Authors possibly mean: Monitoring a test piece with the ACSSM that is fabricated from the same material as the steel structure and exposed to the same environment will contribute to effective decision making for maintenance of the steel structure.
Author Response
Thank you very much for your useful comments and thorough checks. I have entirely revised the manuscript. The paper has been completely revised in accordance with the Reviewer’s suggestions. My responses to the Reviewer’s comments are provided below. The revised parts are indicated by red-colored text in the revised manuscript. I hope that these revisions are satisfactory and the manuscript is acceptable for publication in Metals journal.
The responses in this text are indicated by the red-colored text and the Reviewer’s comments are indicated by the black colored text.
Response to Reviewer 3 Comments
In this manuscript, Purwasih et al. presents the development of an atmospheric corrosion sensor based on strain measurement. The sensor has been constructed by replacing conventional strain gauges with Fiber Bragg grating sensors and by applying the active-dummy method to compensate effects of environmental temperature. The steps of fabricating the sensor and testing sensor’s performance is well-described. Results deserve publication. Authors should take into account the following comments while preparing the final version of the manuscript:
Point 1: I have not received the graphical abstract. If authors have not prepared graphical abstract yet, it should be submitted together with the final version of the manuscript.
Response 1: Thank you for your valuable suggestion. We will submit together with the revised manuscript.
Point 2: References are not following the required ACS style guide. Authors must correct references accordingly (see journal home page), e.g. Journal Articles:
1. Author 1, A.B.; Author 2, C.D. Title of the article. Abbreviated Journal Name Year, Volume, page range.
Response 2: We revised it in the manuscript
Point 3: Line 196: replace “Fig. 6” with “Fig. 7”
Response 3: We revised it in the manuscript
Point 4: The English of the manuscript has to be improved. A final reading is required to remove typing mistakes and correct wrong sentence structures. I name only one example: “Fabricating the test piece from the same material as the steel structures it is monitoring enables the ACSSM to contribute to effective decision making for their maintenance.” Authors possibly mean: Monitoring a test piece with the ACSSM that is fabricated from the same material as the steel structure and exposed to the same environment will contribute to effective decision making for maintenance of the steel structure.
Response 4: Thank you very much for your thorough checks and for giving us an example sentence which is changed with the more clear explanation. We revised it in the manuscript. We will send it to the English editing service after it accepted in this journal.
